# Spinal Stabilization Exercises for Cancer Patients with Spinal Metastases of High Fracture Risk: Feasibility of the DISPO-II Training Program

**DOI:** 10.3390/cancers13020201

**Published:** 2021-01-08

**Authors:** Friederike Rosenberger, Tanja Sprave, Dorothea Clauss, Paula Hoffmann, Thomas Welzel, Jürgen Debus, Harald Rief, Joachim Wiskemann

**Affiliations:** 1Working Group Exercise Oncology, Department of Medical Oncology, National Center for Tumor Diseases (NCT), Heidelberg University Hospital, 69120 Heidelberg, Germany; dorothea.clauss@nct-heidelberg.de (D.C.); paula.hoffmann@nct-heidelberg.de (P.H.); joachim.wiskemann@nct-heidelberg.de (J.W.); 2Department of Health Science, German University of Applied Sciences for Prevention and Health Management, 66123 Saarbrücken, Germany; 3Department of Radiation Oncology, Heidelberg University Hospital, 69120 Heidelberg, Germany; tanja.sprave@uniklinik-freiburg.de (T.S.); thomas.welzel@med.uni-heidelberg.de (T.W.); juergen.debus@med.uni-heidelberg.de (J.D.); 4Department of Radiation Oncology, University Hospital of Freiburg, 79106 Freiburg, Germany; 5Clinical Cooperation Unit Radiation Oncology, German Cancer Research Center (Deutsches Krebsforschungszentrum, DKFZ), and National Center for Tumor Diseases, 69120 Heidelberg, Germany; 6Radiooncological Practice Bad Godesberg, 53177 Bonn, Germany; harald.rief@gmx.at

**Keywords:** advanced cancer, radiotherapy, bone metastases, skeletal metastases, osseous metastases, exercise, training, muscle, strength

## Abstract

**Simple Summary:**

Previous research indicates that the outcomes of irradiation of spinal metastases can be improved through exercise. After this was demonstrated for metastases of low fracture risk, we conducted the first study in patients with spinal metastases of high fracture risk to investigate exercise feasibility. An exercise group performed four spinal stabilization exercises daily over two weeks of radiotherapy, while a control group received relaxation. Patients in the exercise group attended 90% of training sessions, compared to 80% in the control group. No injuries occurred. However, about half of the patients could not perform two out of the four exercises due to pain, weakness or immobility. Nevertheless, they increased exercise time and training-specific strength. Altogether, exercise is well accepted and enhances muscle strength in these patients, but frequent individual adaptations of the training program are needed. This knowledge is a prerequisite for larger studies addressing exercise effects on health.

**Abstract:**

Exercise concomitant to radiotherapy for stable spinal metastases was demonstrated to increase bone density and reduce pain. In the DISPO-II study, the feasibility of exercise concomitant to radiotherapy for unstable spinal metastases was investigated. Here, a detailed analysis of the training program is presented. Cancer patients with spinal metastases (Taneichi score ≥ D) were randomly assigned to an intervention group (INT, *n* = 27, 62 ± 9 years) or control group (CON, *n* = 29, 61 ± 9 years). INT performed spinal stabilization exercises (“all fours”/“plank”/“swimmer”/“band exercise”), and CON received relaxation, daily concomitant to radiotherapy. Exercise attendance rate was 90% in INT and 80% in CON (*p* = 0.126). Within INT, exercise dose increased significantly (*p* < 0.001). 54% of patients could not perform “swimmer” in some or all sessions. 42% could not perform “plank” in some or all sessions. 13 and 25% could not perform “all fours” and “band exercise” in some sessions. “Plank” holding time increased in INT and remained unchanged in CON with different development between groups (*p* = 0.022). Handgrip strength did not develop differently between groups (*p* = 0.397). The exercise intervention demonstrated high acceptability but required frequent modifications due to pain, weakness and immobility to be feasible for the majority of participants. It enhanced specific muscle strength. Larger trials should now investigate exercise effects on health.

## 1. Introduction

Bone metastases are a frequent condition in patients with advanced cancer. About 65 to 75% of patients with metastatic breast or prostate cancer develop bone metastases over the course of their disease, and the spine is often the first bone metastatic site [1]. Clinical challenges include severe pain, several metabolic disturbances, and increased fracture risk [1]. Pathologic fractures not only mean a clinical exacerbation but are also linked to reduced survival and, therefore, need to be avoided [2]. Medical options to reduce fracture risk are osteoprotective medication and, in high-risk patients, prophylactic fixation surgery or palliative-analgetic radiotherapy.

Cancer patients with bone metastases often tend to refrain from physical activity due to safety concerns. However, physical inactivity is associated with a loss in physical function, mobility, balance, protective muscle mass, strength and bone mineral density [3]. This does not only hamper quality of life but also increases fracture risk. Therefore, research needs to provide safe and effective ways of exercising for this patient population.

Until today, five training intervention studies have been published that were specially designed for cancer patients with bone metastases [4,5,6,7,8,9,10,11,12]. Within these studies, two approaches can be distinguished: First, the “no-load approach” avoids load on regions with bone lesions to enable safe exercise for all other parts of the body. Three studies demonstrated its feasibility and positive effects on physical functioning, lower body muscle strength, and quality of life [4,5,6]. Second, the “isometric load approach” specifically trains muscles in regions with bone lesions and, therewith, has the potential to elicit local adaptations. This approach was designed for patients with spinal metastases undergoing palliative-analgetic radiotherapy. Reports from a first study in patients with stable spinal metastases (i.e., low fracture risk, Taneichi score ≤ C [13]) demonstrate that free isometric spinal stabilization exercises reveal positive effects on bone mineral density after radiotherapy, pain, and pain medication use [7,8]. No training-related adverse event, skeletal complication or increased pain occurred in any of these studies.

Encouraged by these findings, an exploratory randomized controlled feasibility trial was conducted to investigate the feasibility of free isometric spinal stabilization exercises concomitant to palliative-analgetic radiotherapy in patients with unstable spinal metastases (i.e., high fracture risk, Taneichi score ≥ D [13]). This DISPO-II study, for the first time, addressed patients with high fracture risk, preexisting fractures and severe bone pain. While general safety/feasibility, as well as secondary clinical endpoints, are published elsewhere [9], we here analyze the feasibility of the training program in detail, including intention-to-treat analysis of attendance, adherence and exercise tolerance as well as training effects on muscle strength. The analysis follows recent recommendations for reporting training attendance and adherence in the exercise oncology setting [14,15]. Knowledge about the training program is a prerequisite for designing larger trials in patients with unstable spinal metastases, powered for effects on bone mineral density, pathologic fractures, and survival.

## 2. Materials and Methods

### 2.1. General Design

The DISPO-II study was an exploratory randomized controlled feasibility trial to investigate the feasibility of spinal stabilization exercises concomitant to palliative-analgetic radiotherapy in cancer patients with unstable spinal metastases (clinical trials no. NCT02847754). The intervention group (INT) received supervised free spinal stabilization exercises (i.e., exercises without training-machines) on each day of radiotherapy (~2 weeks) and continued the training program home-based and unsupervised for 3 months. The social-interaction control group (CON) received muscle relaxation training in the same way. The primary endpoint was general feasibility, expressed as training attendance rate and training-related adverse events [16] until 3 months after the end of radiotherapy, and is published elsewhere [9]. Here, we focus on the supervised intervention from baseline to the end of radiotherapy and (a) report intention-to-treat analysis of training feasibility, i.e., attendance and adherence metrics including completed exercise dose and exercise tolerance according to recently published recommendations [14,15] and (b) present training effects on muscle strength.

### 2.2. Participants

Recruitment took place at the Heidelberg Institute of Radiation Oncology (HIRO), Heidelberg University Hospital. Sixty patients with histologically confirmed cancer and one or more unstable spinal metastases in the thoracic or lumbar spine or os sacrum with an indication for palliative-analgetic radiotherapy were included in the study. The stability of the spinal metastases was rated using the Taneichi score [13]. It was derived from computer tomography (CT) scans and takes into account the size and location of the metastasis within the vertebral body. A score of ≥D is considered unstable and served as an inclusion criterion. Further inclusion criteria were [17]: age between 18 and 80 years; Karnofsky performance index ≥70; Bisphosphonate or anti-RANK ligand antibody therapy; and signed informed consent. Exclusion criteria were [17]: fixation surgery for the unstable spinal metastases; significant neurological, psychiatric or other disorders or medical or psychological conditions that may prevent participation, completion or understanding of the study or study consent; and obvious inability to perform the exercises (at the point of inclusion). Participants were randomized 1:1 to INT or CON.

### 2.3. Interventions

INT received isometric spinal stabilization training daily for 10 ± 2 days (min–max: 5–17 days) concomitant to radiotherapy. Training was supervised 1:1 by an exercise physiologist or physiotherapist and was performed without a corset. The program took around 15 min per day and consisted of four free spinal stabilization exercises that are displayed in Figure 1 (see also Appendix A). Holding time in each position started with 20 s and increased from session-to-session when possible by as much as movements still appeared safe. In the first session, the exercises were explained based on pictures and demonstrated practically. Special emphasis was given on keeping the spine straight during all movements, including going down to the floor and up again, and not to turn towards the trainer when talking during exercise.

To go down to the “all fours” position (Figure 1a), the patient stood opposite of the trainer, took a one-leg step towards him, held his hands for assistance and went down to one knee and the other. Then, the upper body came down to place the hands on the floor. “All fours” were performed with each extremity stretched separately. To go to the “plank” position (Figure 1b), the patient moved his hands forward in small steps and then came down to the forearms. Then, he got on his tiptoes and lifted the knees from the floor. A hip position slightly >0° was targeted to avoid sagging. To lay down for the “swimmer” position (Figure 1c), the patient moved his shoulders forwards/downwards by flexing his elbows. The “swimmer” was performed with toes on the floor and arms elevated to the side (bent arms) or front (straight arms) depending on the patient’s mobility. From here, the patient came back to “all fours” and stood up as he went down. The trainer assisted in the first sessions and taught the patient how to use a chair for assistance in the last sessions. For the “shoulder blade band exercise” (Figure 1d), the patient put the band under tension in front of the trunk by pulling the shoulder blades together and varied the height of the band from the belly to the shoulders while maintaining the tension of the band. If a patient was unable to perform an exercise, “band rowing” was provided as a substitute. This was performed in a standing position with the patient starting with straight arms in front of the body and pulling the elbows backward and the shoulder blades together to hold this position.

CON received progressive muscle relaxation exercises daily on 9 ± 2 days (min–max: 5–11 days) concomitant to radiotherapy. Progressive muscle relaxation instructions were either read or played from recordings by study personnel up to patient choice. The program took around 15 min per day and consisted of exercises for the upper and lower extremity, but not for the back to avoid training effects. It was performed in either supine or sitting position depending on the patient’s mobility.

### 2.4. Testing Procedures

Strength testing was performed in the first and the last training session. As a training specific strength test, maximum “plank” holding time was assessed. Patients were instructed to hold the position as long as possible, and time was taken. Estimated hip angle was noted to enhance standardization. As an unspecific strength test, the handgrip strength of the dominant site was assessed using a Jamar handgrip dynamometer (Patterson Medical, Warrenville, IL, USA). The test was performed prior to exercise in a sitting position with the elbow attached to the trunk and an elbow angle of 70 to 90°. The best out of three attempts was analyzed.

### 2.5. Analysis of the Training Program

Attendance rate was quantified for INT and CON as lost to follow-up and percentage of completed sessions. Further analyses were performed within INT. The percentage of patients was analyzed who missed single training sessions (defined as up to 2 consecutive sessions), interrupted training for ≥3 consecutive sessions, and discontinued training permanently. Adherence metrics included the completed cumulative exercise dose, which was calculated for each session as the sum of exercise holding time in seconds (i.e., total holding time of “all fours” plus “plank” plus “swimmer” plus “shoulder blade band exercise”). It is presented for the whole group and single cases to highlight variability. To quantify dose modifications, percentages of patients and sessions with an increase and decrease in holding time of single exercises were analyzed. Furthermore, to analyze exercise tolerance, for each of the four exercises, the percentage of patients who were unable to perform it or needed to modify it was analyzed. Altogether, the analysis followed recent recommendations, which were adapted for free exercises instead of endurance or machine-based resistance training [14,15].

### 2.6. Statistical Analysis

Data were analyzed using IBM SPSS Statistics 25 for Microsoft (IBM, Armonk, NY, USA). Nominal data are presented as frequencies, metric data are presented as means ± standard deviations (SD) when normally distributed or medians and quartiles (Q1; Q2) when skewed. Differences in baseline characteristics between groups were tested using chi-squared tests for nominal data and independent samples’ *t*-tests for metric data. Changes in the completed cumulative exercise dose from the first to the last training session in INT were tested using paired samples’ *t*-tests. Changes in handgrip strength from baseline to follow-up were compared between groups using a two-way repeated-measures ANOVA (interaction effect and main effects for time and group). Changes in “plank” holding time from baseline to follow-up were, because of skewed data, analyzed using a Mann–Whitney test for between-group differences of the change score (i.e., change from baseline to follow-up) and Wilcoxon tests for within-group comparisons.

## 3. Results

### 3.1. Patient Characteristics

Out of the 60 included patients, 27 patients in INT and 29 patients in CON started the study intervention and were included in the analysis. A participant flow chart is given in Figure 2. Patient characteristics are given in Table 1. There were no differences between groups except for the spinal instability neoplastic score (SINS) [18], which was significantly higher (indicating higher instability) in INT.

### 3.2. Attendance Metrics.

In INT, twenty-seven patients started the study intervention, and three were lost to follow-up due to poor health. In CON, twenty-nine patients started the study intervention, and two were lost to follow-up because of unwillingness to follow the study protocol (Figure 2). Median and quartiles of training attendance rate were 9(7; 10) sessions (90(70; 100)%) in INT and 6(2; 9) sessions (80(21; 95)%) in CON with no significant differences between groups (*p* = 0.216). No training-associated adverse event or skeletal complication occurred in either group.

In INT, fifteen patients (56%) missed single training sessions, two (7%) interrupted the training program for ≥3 consecutive sessions, and four (15%) discontinued the training program. Of these, there were those three patients mentioned above who were lost to follow-up who completed 5, 0, and 1 training session as well as one more patient who completed 1 training session but participated in the follow-up test. In total, 59 out of 271 possible training sessions (22%) were missed and reasons were health-related (pain, poor health, weakness: *n* = 46 sessions), organizational (*n* = 6 sessions), or unknown (*n* = 7 sessions).

### 3.3. Adherence Metrics

The completed cumulative exercise dose in INT increased from 169 ± 74 to 358 ± 200 s from the first to the last training session (*p* < 0.001) in those patients who performed at least 2 training sessions (*n* = 24). Single courses over time are displayed in Figure 3. Exercise dose for single exercises was increased in all twenty-four patients (100%) and a median of 7(4; 8) sessions (73(55; 79)%) in the sense of training progression as planned. However, it was also reduced in twenty-two (92%) of patients and a median of 2(1; 3) sessions (28(13; 55)%) due to pain, poor health or weakness.

Out of the four exercises, “swimmer” required the most modification: It could not be performed by thirteen patients (54%) in a median of 67(39; 100)% of sessions and needed to be modified (one arm could not be raised) in one patient (4%) in all sessions due to pain and immobility. “Plank” could not be performed by ten patients (42%) in a median of 55(13; 100)% of sessions and needed to be modified (knees kept on the floor) in one patient (4%) in 55% of sessions due to pain. “All fours” could not be performed by three patients (13%) in a median of 67(33; 67)% of sessions and needed to be modified (raising of arms or legs only) in four patients (17%) in a median of 36(10; 90)% of sessions due to pain, immobility or weakness. “Shoulder blade band exercise” could not be performed in six patients (25%) in 27(13; 55)% of sessions due to pain or weakness, which occurred most often in the last out of the four exercises. Skipped exercises were replaced by “band rowing” in twelve patients (50%) in 74(46; 100)% of sessions.

### 3.4. Changes in Muscle Strength

Of those patients who completed the baseline and follow-up tests, “plank” holding time tests were not possible due to immobility and pain in six cases (24%) in INT and in nine cases (33%) in CON. Handgrip tests were missing due to organizational failure in two patients (7%) in CON. Strength test data are given in Table 2. There were no baseline differences between groups (“plank” holding time: *p* = 0.825, handgrip strength: *p* = 0.399). “Plank” holding time developed significantly different between groups (*p* = 0.022). It increased in INT by a median change score of 16(0; 40) s (*p* = 0.002) and remained unchanged in CON (2(−2; 11) s, *p* = 0.185). Handgrip strength did not develop differently between groups (*p* = 0.397) and did not change within either group (INT −0.1 ± 2.1 kg, CON −0.7 ± 2.7 kg, *p* = 0.231).

## 4. Discussion

In the DISPO-II study, free isometric spinal stabilization exercises were applied in cancer patients with unstable spinal metastases (Taneichi score ≥ D [13]) undergoing palliative analgetic radiotherapy. Attendance metrics were overall high, indicating high acceptance of the training program. However, adherence analysis revealed frequent deviations from the training prescription in a way that single exercises could not be performed or needed to be modified, mainly due to pain, immobility or weakness. Furthermore, exercise dose had to be reduced in single sessions. Nevertheless, exercise dose and training specific muscle strength significantly increased over time on average, demonstrating the efficacy of the training program.

Previous studies especially designed for cancer patients with bone metastases either excluded patients with unstable spinal metastases [7,8] or avoided load on regions with bone lesions [4,5,6]. Other studies were not specially designed for patients with bone metastases but included some, among others. Most of these studies defined painful or unstable bone metastases as exclusion criteria [19,20,21,22,23,24,25,26], medical clearance as a prerequisite [27,28,29,30] and/or modified the training program to enhance safety [29,30,31]. Only three studies reported no special attention when including patients with bone metastases [32,33,34]. Out of these studies in mixed populations, only soccer intervention studies reported exercise-related bone events, but these were not associated with metastatic sites [22,23,24,27,28]. Altogether, given precautions and modifications, exercise appears safe and feasible in patients with stable bone metastases. Beyond that, the present exploratory randomized feasibility trial for the first time suggests that free spinal stabilization exercises are safe (in terms of exercise-related adverse events or skeletal complications) and given frequent individual adaptations also feasible even in a more vulnerable population of patients with advanced cancer who undergo palliative-analgetic radiotherapy for unstable spinal metastases and partly present with severe pain, pain medication use, previous fractures, corset, and limited life expectancy.

In the present analysis, recent recommendations for reporting training attendance and adherence in the exercise oncology setting [14,15] were for the first time transferred from endurance and machine-based resistance training to free exercises. Attendance and adherence metrics, including cumulative exercise dose and dose modifications, proved applicable and informative. Cumulative exercise dose was not calculated as the sum of sets × repetitions × load, but as the sum of exercise holding time, which also gives an idea of training progress since the weight of free exercises remains relatively constant. However, relative dose intensity (RDI), defined as the ratio of total completed to total prescribed exercise dose [14], could not be calculated. In addition to the recommended metrics, the percentage of patients who were unable to perform an exercise or needed to modify an exercise was reported for each exercise and session. This appears informative in settings where the feasibility of single exercises is questionable.

Regarding the single exercises, “all fours” and “shoulder blade band exercise” turned out feasible, with ≥75% of patients being able to perform it in all training sessions. In contrast, the feasibility of “swimmer” and “plank” was poor, with 54 and 42% of patients being unable to perform it in some or all sessions. Both exercises require going down on the floor and up again, which is uncomfortable for individuals with immobility and pain. However, these exercises are expected to be highly effective, and patients reported that they regained confidence by being supposed to do them. Therefore, it appears appropriate to strive for the training plan as is and be prepared for frequent individual adaptations. These adaptations and the supervision, in general, require sensitivity because the common rule of “pain-free exercise” is not applicable in a population with high pain levels and pain medication use. Therefore, 1:1 supervision by a qualified exercise expert is crucial, at least in the beginning.

From a practical point of view, it would make sense to add strengthening exercises for the thighs to the training program to improve the ability and safety of going down to the floor and up again. Squats and stationary lunges with assistance appear possible as long as no unstable bone metastases in the femur or acetabulum exist. Furthermore, isolated exercises for the upper extremity may be possible, given the absence of unstable metastases located there. This means that the DISPO-II exercises may be combined with the initially mentioned “no-load approach” to enable whole-body strengthening. However, exercise should not take too long in total, bearing in mind that the last out of the four DISPO-II exercises was missed in one-quarter of patients in several sessions due to accumulating weakness and pain.

The question arises whether, based on this exploratory feasibility trial, the DISPO-II exercises should be recommended to the studied population. The exercises were found to be safe and given individual adaptations also feasible and effective in terms of specific muscle strength. However, regarding the quality of life, pain, pain medication use, bone density and pathologic fractures, the study revealed no effects [9]. Although this is attributable to the small sample size (the study was not powered for these endpoints), proof of positive effects is so far missing. Until larger studies clarify this, the DISPO-II training program may be offered to patients who are intrinsically motivated to exercise and ask for options. However, the data do not justify general exercise recommendations for health purposes.

The major limitation of this exploratory controlled feasibility trial is its small sample size. This is especially important for the interpretation of safety data. A lot more than 24 patients need to complete the exercise program to consider it finally safe. In terms of feasibility and effects on muscle strength, the small sample size appears less relevant. The randomized controlled design allows the conclusion that the attendance rate was similar in INT and CON. There may have even been a disadvantage for INT because the exercises needed to be performed on the floor while CON was allowed to sit for relaxation training when laying down was difficult or painful. Adherence turned out to be highly variable. This is unlikely to change in a larger sample. Finally, changes in training specific muscle strength were found to be significant even in this small group of patients.

## 5. Conclusions

In the present exploratory controlled feasibility trial, free spinal stabilization exercises were administered to cancer patients with unstable spinal metastases (Taneichi score ≥D [13]) concomitant to palliative-analgetic radiotherapy by qualified exercise specialists. Analyses of the training program suggest that the DISPO-II exercises are well accepted. Given frequent adaptations due to pain, weakness and immobility, they are also feasible for the majority of participants and effective to enhance training specific muscle strength. Against this backdrop, larger trials should be performed to confirm safety and investigate training effects on clinical outcomes like bone mineral density, pathologic fractures, and survival.

## Figures and Tables

**Figure 1 cancers-13-00201-f001:**
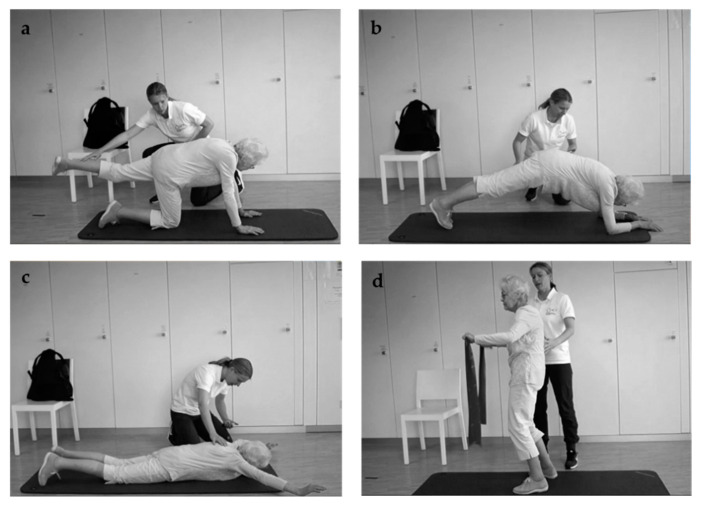
Exercises of the DISPO-II study: (**a**) “all fours”, (**b**) “plank”, (**c**) “swimmer”, (**d**) “shoulder blade band exercise”.

**Figure 2 cancers-13-00201-f002:**
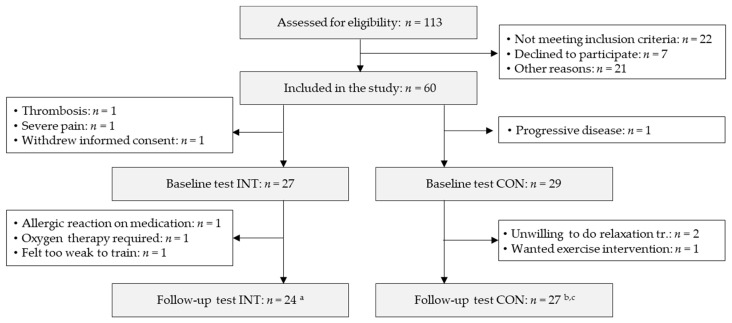
Participant flow chart from baseline (start of radiotherapy) to follow-up (end of radiotherapy). INT: intervention group, CON: control group, missing data: ^a^ “plank” holding time test INT: *n* = 18, ^b^ “plank” holding time test CON: *n* = 18, ^c^ handgrip test CON: *n* = 25.

**Figure 3 cancers-13-00201-f003:**
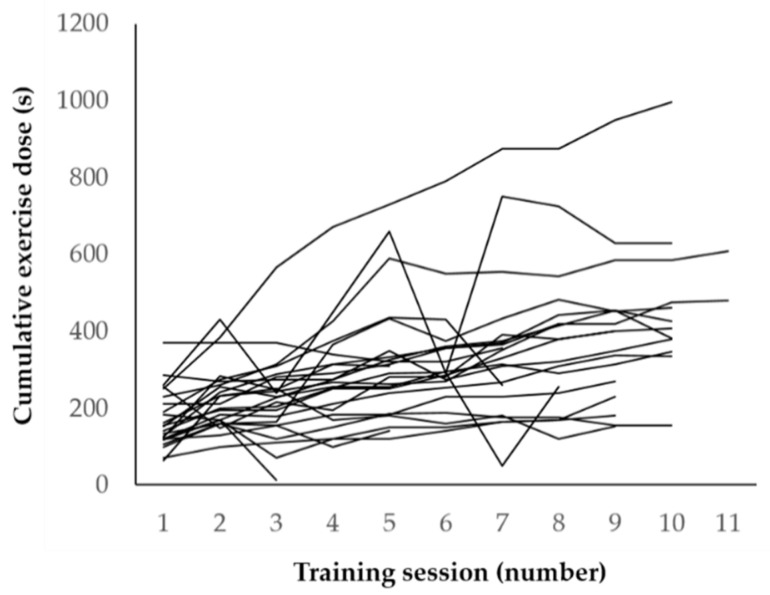
Single courses of the completed cumulative exercise dose over time.

**Table 1 cancers-13-00201-t001:** Patient baseline characteristics.

Characteristics		INT (*n* = 27)	CON (*n* = 29)	*p*
Anthropometric data				
	Males, *n* (%)	13 (48)	12 (41)	0.611
	Females, *n* (%)	14 (52)	17 (59)	
	Age (years), mean ± SD	62 ± 9	61 ± 9	0.657
	Height (cm), mean ± SD	172 ± 9	170 ± 9	0.497
	Weight (kg), mean ± SD	72 ± 11	75 ± 15	0.301
	BMI (kg/m^2^), mean ± SD	24.4 ± 4.1	25.8 ± 4.6	0.158
Cancer type				
	Lung cancer, *n* (%)	8 (30)	14 (48)	0.654
	Breast cancer, *n* (%)	7 (26)	6 (21)	
	Prostate cancer, *n* (%)	4 (15)	2 (7)	
	Other, *n* (%)	6 (29)	7 (24)	
Irradiated spinal metastasis				
	Thoracic spine, *n* (%)	20 (74)	24 (83)	0.429
	Lumbar spine, *n* (%)	7 (26)	5 (17)	
	Radiotherapy fractions, mean ± SD	10 ± 2	9 ± 2	0.690
	Inpatient stay for radiotherapy, *n* (%)	13 (48)	11 (38)	0.440
Characteristics of the spinal metastasis				
	Osteolytic, *n* (%)	10 (37)	15 (52)	0.227
	Mixed, *n* (%)	15 (56)	14 (48)	
	Osteoblastic, *n* (%)	2 (7)	0 (0)	
	SINS score, mean ± SD	12.0 ± 2.5	10.3 ± 2.2	0.010
	Mizumoto score, mean ± SD	5.0 ± 2.0	5.5 ± 1.7	0.298
	Pathologic fracture, *n* (%)	17 (63)	11 (38)	0.061
	Orthopedic corset, *n* (%)	10 (37)	14 (48)	0.396
Other metastatic sites				
	Other bone metastases, *n* (%)	27 (100)	27 (93)	0.156
	Brain metastases, *n* (%)	7 (26)	5 (17)	0.429
	Lung metastases, *n* (%)	5 (19)	11 (38)	0.108
	Visceral metastases, *n* (%)	9 (33)	13 (45)	0.379
	Soft tissue metastases, *n* (%)	2 (7)	7 (24)	0.088
Medication				
	Opiate, *n* (%)	15 (56)	16 (55)	0.977
	NSAID, *n* (%)	20 (74)	22 (76)	0.877
	Dexamethasone, *n* (%)	5 (19)	2 (7)	0.189
	Psychiatric medication, *n* (%)	8 (30)	8 (28)	0.866
	Sleeping medication, *n* (%)	4 (15)	9 (31)	0.151
Life expectancy				
	Died within 3 months from baseline, *n* (%)	11 (41)	6 (25) ^a^	0.243
	Died between 3 and 6 from baseline, *n* (%)	4 (15)	4 (17) ^a^	0.894

INT: intervention group, CON: control group, BMI: body mass index, NSAID: nonsteroidal anti-inflammatory drugs, ^a^ survival data of five control group patients were unavailable.

**Table 2 cancers-13-00201-t002:** Handgrip strength (mean ± SD, INT: intervention group, *n* = 24, CON: control group, *n* = 25) and “plank” holding time (median and quartiles, INT: *n* = 18, CON: *n* = 18) at baseline and follow-up (* significant within-group difference from baseline to follow-up).

Strength Test	INT		CON		*p*
	Baseline	Follow-Up	Baseline	Follow-Up	
Handgrip strength (kg)	28.0 ± 8.4	27.9 ± 8.7	26.0 ± 7.9	25.3 ± 8.1	0.397
“Plank” holding time (s)	19 (10;51)	31 (18;100) *	20 (11;51)	29 (14;46)	0.022

## Data Availability

The data presented in this study are available on request from the corresponding author.

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
