# Peer review of "Spinal Stabilization Exercises for Cancer Patients with Spinal Metastases of High Fracture Risk: Feasibility of the DISPO-II Training Program"

_cancers, 2021, doi:10.3390/cancers13020201_

Round 1

Reviewer 1 Report

General comments

Overall a very well-written and important study. However, it does only focus on a small part of a previously published version of this study. It does offer one new endpoint (strength) and reports additional information on feasibility and uses intention-to-treat analysis. However, these really should have been included in the first publication to comply with best practice.

The other additional aim was to report training details (which are missing from the previously published version), but these are reported in the published trial protocol.

The previously published paper of the same trial (Paravertebral muscle training in patients with unstable spinal metastases receiving palliative radiotherapy: an exploratory randomized feasibility trial) states the intervention was feasible ‘with a clear majority of patients being able to complete the assigned regimen’. This seems a bit misleading now since your discussion in this manuscript states that feasibility of 2 of the 4 exercises were poor. Nevertheless, the paper does provide important insight into the true feasibility of these exercises.  

Specific comments

Abstract

Conclusion: It would be more correct to state that an adaptation of the exercises was feasible as 2 of the 4 exercises themselves don’t appear to be that feasible. Simply stating ‘When being prepared for frequent adaptation…’ is a bit vague.

Perhaps: ‘The intervention demonstrated high acceptability but required frequent modification to be feasible for the majority of participants’.

Introduction:

Please clarify the isometric exercises in the introduction – references don’t seem to match up e.g. 5 references to support the statement ‘In a first study…’ which implies only 1 study. Perhaps this should read ‘In a series of studies by the same group’ or ‘Reports from one randomized controlled trial…’ (otherwise it looks like many different trials have been conducted). Consider only citing the main paper/s you refer to (RE: pain and bone mineral density) rather than all your papers. 

Method:

The participant section jumps straight to results. Please briefly cite eligibility criteria and recruitment here (and refer to protocol) and put the actual participant numbers and flow in the results section.

Intervention:

Good, clear explanation making replication possible.

Discussion:

Careful and considered discussion

Reviewer 2 Report

General comments

This is a highly unique and clinically interesting exercise protocol that aims to detail the exercise program trialed, feasibility of the program and effects on muscle strength in people with unstable skeletal metastases. This study is very interesting because it goes into great detail in a clinical population rarely studied. However, I have some questions and concerns relating to the conclusions drawn by the research team.

Specific comments

  • Line 26: “Exercise proved safe”. Could you please provide the reader with a clearer picture of what this statement relates to. I understand that it is the simple summary but something to reflect that there were no adverse events in the exercises that the patients did perform. We do not know if the patients had done all the prescribed exercises as each session if this program would have been safe.
  • Line 27: If this is in relation to the INT exercise group could you please add this detail in to this sentence.
  • Line 39: “In INT, exercise dose increased significantly (p<0.001).” Please clarify what this means? Do you mean that they attended more sessions week 1 to week 2? Or do you mean that they were doing significantly more exercise (general PA? or study exercises) than the control group were doing the relaxation exercises? Or is this a within groups comparison? Please clarify.
  • Line 30: “Swimmer” could not be performed by 54% of patients in 67% of sessions. “Plank” could not be performed by 42% in 55% of sessions. “All fours” and “band exercise” could not be performed by 13 and 25% in 67 and 27% of sessions”. I understand that this is tricky information to communicate, but I think it needs more detail. For example, with Swimmer- does this 67% related to all sessions or just 67% of the sessions of these 54% of the participants?
  • Line 42: I understand that this is a within groups comparison, is there a reason you have not presented the between groups comparisons give that the between groups comparison is the main analysis of this RCT?
  • Line 43: Which group and/or analysis is this referring to? Ex or control? Between or within groups analysis?
  • Line 91: what is meant by “free” when you mention free spinal stabilization exercises? Please detail. When I first read the sentence, I read it as cost free, but realized later in the paper that is perhaps not what you meant?
  • Section 2.5: Are the authors using attendance and adherence interchangeably? Generally, they are two difference aspects of monitoring an exercise program, attendance: did the person turn up to the session and adherence: did the person do the exercises once they attended the session. How are these terms used in this paper? Please clarify and consider using descriptors. For example, it seems in the paper that adherence could be referring to adhering to the trial protocol (in sessions number but also later in the results you report exercise adherence) rather than just to the specific exercises included. Please add more detail.
  • Statistical analysis:

My main concern of the paper: I understand that you used Mann-Whitney U tests for the between groups analysis of the plank times at follow-up. Were differences between groups analysed at baseline? My main concern is that, without checking for any differences at baseline, how can one know for sure that any differences (or no differences) one is seeing at followup, is not a result of differences at baseline. It is problematic therefor to draw conclusions on this outcome without knowing about baseline. Could you log transfer the skewed data? There are a few ways to overcome this problem (you could also just simply check for differences at baseline for this outcome) but without taking into consideration potential baseline differences, I do not feel that this conclusion about plank time can be made with good confidence. I also understand the within groups comparisons may be of interest in terms of absolute values (which are also very small in the study, just a few seconds average between the groups I see in Table 2 31s v 29 sec) and change, but I would put forward that when conducting an RCT that the main analysis should always be the between groups analysis.

  • Results: Table 2: I am confused with the differences between the numbers in the text above table 2 and table 2 in regard to plank times. Why are these numbers different? You say the INT group improved by 16 sec but in the Table, it seems that the improvement is from 19 to 31 sec= 12 sec. and similar with CON median changing from 20 sec to 29 sec = 9 sec. I am surely missing something but could the research team please clarify the differences. Are the data from different subgroups of the study population?
  • Conclusions: As mentioned above, it is difficult with good confidence to agree with the authors that the exercises were effective to enhance training specific muscle strength (ie plank time)

Reviewer 3 Report

General comments:

The aims of this study are based on an exploratory randomized controlled trial called DISPO-II intended to examine the effect of exercise intervention in patients with spinal metastases of high fracture risk and to check the feasibility of this training program. Exercise intervention has been compared to a control group with relaxation. Based on the literature, the authors identified two types of intervention (no load vs isometric) in patients with spinal metastases that seem to have effects on different aspects of the disease. Overall, the study is well written and interesting, but some information is lacking to fully understand the ins and outs of this study.

In the introduction section, it might be necessary to explain why the authors focused more on the isometric load approach. In addition, the authors wrote that the feasibility of the exercise intervention had already been published but added other information such as training prescription, intention-to-treat analysis of adherence and exercise tolerance as well as the effects of training on muscle strength. It could be interesting to report in a few words what the previous study has shown in terms of feasibility otherwise, it does not seem necessary to talk about this study.

In the method section (general design), the authors wrote that this study is (in part) aimed at examining the effect of the training on the muscle strength. However, isometric program seems to be efficacious on bone mineral density after radiotherapy, pain, and pain medication use. Why the authors are expecting change on muscle strength? In addition, in the same section, it might be necessary to clarify whether the control group had to also make relaxation in a home-based design, during three months after the end of the radiotherapy. How were supervised and controlled the home-based sessions?

In the intervention presentation, the authors wrote “Holding time in each position started with 20 s and increased from session to session when possible”. Some more explanation on the increased difficulty is required. One aim of the study is to give information on the training prescription, so this information is needed. In addition, the authors wrote that relaxation was performed in either supine or sitting position depending on the patient’s mobility. What bothers me here is that the possibility of being seated was considered whereas for the INT group most of the exercises are done on the floor. I think that this aspect might be considered as a limitation of the study. How was planned the progression in relaxation?

In the analysis of the training program section, I was wondering if the authors measured adherence or compliance. I think it is rather compliance because the authors are focused on the respect of the prescription. Moreover, nothing is said on the home-based session? How were the session scheduled and controlled?

In the results section, What is the SINS’ score? It is not clear whether the patient had a minimum of sessions per week? Minimum and maximum sessions should be reported. The patients had 3 months and 15 days of training. How was calculated 271 possible sessions training? Clear information is required. The adherence metrics section needs more details because it is not easy to understand how the completed cumulative exercise dose increased. How was measured the exercise dose?

In the discussion section, the authors wrote “the present exploratory randomized feasibility trial for the first time suggests that free spinal stabilization exercises are safe and feasible.” I am not sure that the authors can make this conclusion because pain was a major barrier not to exercise. For some of the patients, I am not sure that the intervention was assessed as safe.

Minor comments

Some description of the Taneichi score would be good.

The comments under the figure 1 are difficult to understand, especially the test measure (a, b, c). Does it mean that the INT group did not make the handgrip?

Round 2

Reviewer 3 Report

The authors took most of the suggestions into account and answered the questions clearly. The other corrections made (other reviews) also improved the quality of the manuscript. The information needed for a replication study is available in the manuscript.